# Study on the Impact of Building Energy Predictions Considering Weather Errors of Neighboring Weather Stations

**DOI:** 10.3390/s24041157

**Published:** 2024-02-09

**Authors:** Guannan Li, Yong Wang, Chunzhi Zhang, Chengliang Xu, Lei Zhan

**Affiliations:** 1School of Urban Construction, Wuhan University of Science and Technology, Wuhan 430065, China; guannanli@wust.edu.cn (G.L.); wangyong_stu@163.com (Y.W.); xcl120@126.com (C.X.); zl0224760606@163.com (L.Z.); 2Anhui Province Key Laboratory of Intelligent Building and Building Energy-Saving, Anhui Jianzhu University, Hefei 230601, China; 3Key Laboratory of Low-Grade Energy Utilization Technologies and Systems (Chongqing University), Ministry of Education of China, Chongqing University, Chongqing 400044, China; 4State Key Laboratory of Green Building in Western China, Xi’an University of Architecture & Technology, Xi’an 710055, China

**Keywords:** building energy predictions, weather errors at neighboring weather stations, long short-term memory (LSTM), predicting next 1 h vs. predicting next 1 day

## Abstract

Weather data errors affect energy management by influencing the accuracy of building energy predictions. This study presents a long short-term memory (LSTM) prediction model based on the “Energy Detective” dataset (Shanghai, China) and neighboring weather station data. The study analyzes the errors of different weather data sources (Detective and A) at the same latitude and longitude. Subsequently, it discusses the effects of weather errors from neighboring weather stations (Detective, A, B, C, and D) on energy forecasts for the next hour and day including the selection process for neighboring weather stations. Furthermore, it compares the forecast results for summer and autumn. The findings indicate a correlation between weather errors from neighboring weather stations and energy consumption. The median R-Square for predicting the next hour reached 0.95. The model’s predictions for the next day exhibit a higher Prediction Interval Mean Width (139.0 in summer and 146.1 in autumn), indicating a greater uncertainty.

## 1. Introduction

### 1.1. The Importance of Building Energy Predictions and Their Uncertainty

In the era of big data, which emphasizes energy saving and emission reduction, exploring a more rational energy deployment for energy demand and identifying areas of improvement by predicting future building energy is an important approach in the quest for energy-saving solutions in the building sector [1,2]. However, the prediction of building energy is challenging and inherently uncertain, encompassing two distinct categories [3]: arbitrary and epistemic. Arbitrary uncertainty arises from stochastic or natural processes associated with the system under study, such as weather variables, while epistemic uncertainty is due to a lack of knowledge and can be reduced by acquiring additional data [3,4]. Cognitive uncertainty, such as uncertainty regarding the parameters of the prediction model itself, presents a significant challenge for accurate prediction.

### 1.2. Factors Contributing to the Sources of Uncertainty in Building Energy Predictions

Since building energy consumption is influenced by various factors, such as weather conditions and building characteristics, predicting building energy consumption faces several challenges. Among these factors, weather conditions play a crucial role and cannot be ignored in building energy prediction. Traditional building energy prediction models tend to focus only on internal building characteristics, disregarding the impact of the external environment. However, weather data, which are typically obtained from weather stations, may have limited availability and accessibility for retrieval. As weather data are now essential inputs for energy forecasts, the errors in these data deserve greater attention. Many scholars have studied weather data errors, including measurement errors [5,6,7], sensor drift [8,9], missing data [10], noisy data [2,11], and others. For instance, Wouter et al. [7] analyzed error models for weather forecasting using historical observations and weather forecast data. Arx et al. [9] detected and corrected sensor drift in long-term weather data. Impedovo et al. [10] proposed an effective machine learning solution for real-time weather parameter prediction in scenarios with missing data. Arnab et al. [11] developed a Veracity Score (VS)-based method for analyzing noisy spatial data and evaluated it through extensive simulations. Building upon these types of errors, this study aims to analyze their impact. Additionally, most public weather stations are typically sparsely located in specific areas, such as airports. Due to complex topographical features or the urban heat island effect, weather conditions in urban areas can differ significantly from those at the building’s location. In practice, weather data from nearby weather stations are often used as a reference for predicting a building’s energy consumption. However, this simple reference can lead to prediction errors due to factors such as the distance between the weather station and the building, as well as differences in terrain. Researchers have utilized weather data from publicly available sources, including airports [12], local weather stations [13], or online weather forecasting websites [14]. These weather data usually represent an entire city, region, or even a larger area, which may result in greater prediction errors due to the relative distance between the weather data collection location and the target Model Predictive Control (MPC) building. Therefore, it is important to consider weather errors from neighboring weather stations for accurate building energy predictions.

### 1.3. Overview of Building Energy Prediction Methods

Methods used for building energy predictions are typically categorized into two main approaches: physical methods [15] and data-driven methods [16,17,18]. Physical methods require detailed information about the building energy system for modelling purposes. On the other hand, the data-driven approaches are emerging as analytical techniques that transform data into information, enabling a shift from perceptual to rational understanding. In practice, data-driven approaches are often more feasible, flexible, and accurate compared to physical model-based approaches [19,20]. In recent years, artificial intelligence (AI) [21] has gained wide usage in various fields, including autonomous driving [22] and robot manipulator dynamics [23]. AI-based techniques are primarily data-driven, as they analyze existing observational data and establish input–output mapping relationships to predict specific building energy data. With advancements in computing power and algorithms, numerous data analysis and mining techniques have rapidly developed. These data-driven techniques for building energy prediction encompass shallow machine learning methods [24,25], such as support vector machines (SVMs) [26,27,28]. Traditional machine learning-based prediction methods, like SVMs, are capable of handling nonlinear problems and generating accurate predictions. For instance, Li et al. [29] compared SVMs and artificial neural networks (ANNs) and found that SVMs outperformed ANNs. Guo et al. [30] compared multivariate linear regression, SVMs, ANNs, and extreme learning machines, concluding that extreme learning machines yielded the best results. Similarly, Fan et al. [31] compared seven machine learning algorithms, including multivariate linear regression, resilient networks, random forests, gradient augmentation machines, support vector machines, extreme gradient boosting, and deep neural networks. They found that extreme gradient boosting (XGBoost) combined with deep auto-coding performed the best. It is important to note that these methods may have limitations in modelling short-term dependencies and could require more computational resources and time when dealing with complex problems. Traditional time series forecasting methods, for example, may focus more on modelling short-term dependencies, potentially resulting in poor predictions for long-term trends and complex patterns. To address this issue, Wang et al. [2] compared twelve load forecasting methods, including three heuristics, seven shallow machine learning algorithms, and two deep learning algorithms. XGBoost provided the most accurate predictions in the shallow machine learning category, while Long Short-Term Memory Network (LSTM) performed best in the deep learning category. XGBoost demonstrated superior performance in long-term forecasting, whereas LSTM outperformed in short-term forecasting and exhibited greater robustness to input uncertainty. Additionally, Wang et al. [32] compared LSTM and autoregressive integrated moving-average (ARIMA) models and found that LSTM outperformed ARIMA in plug load prediction.

Considering the regularity of energy consumption sequence data, LSTM models have made significant progress in the field of building energy prediction, leading to improved model accuracy [33]. LSTM models, being powerful sequence models, have garnered increasing attention from researchers due to their ability to capture nonlinear relationships. Several studies have demonstrated the superiority of LSTM models in building energy prediction tasks, especially in capturing long-term dependencies in time-series data. For instance, Wang et al. [2] employed LSTM to predict the energy consumption of university buildings, including offices and laboratories, showcasing the superior short-term prediction performance of LSTM compared to traditional methods. In a similar vein, Zhou et al. [34] utilized LSTM to predict the energy consumption of library air conditioning systems. Accurately predicting energy levels in future time periods not only aids building managers in optimizing energy usage and reducing operating costs but also provides decision support to mitigate adverse environmental impacts. In other domains, such as power systems, accurate load demand forecasts at different time scales (general, short-term, medium-term, and long-term) are essential for the proper operation of power facilities [35,36,37]. For instance, economic dispatch of generation capacity, fuel purchasing dispatch, security analysis, and short-term maintenance scheduling require short-term forecasts ranging from hours to weeks in advance. Real-time control and real-time security assessment necessitate short-term forecasts from minutes to hours ahead [38]. In the context of building energy prediction, employing LSTM models to predict energy consumption for the next hour and the next day (i.e., 24 h) is crucial for gaining a comprehensive understanding of future trends and making informed energy management decisions.

### 1.4. Content of This Study

In this study, an energy prediction model was constructed using the “Energy Detective” dataset (Shanghai) [39] and data from a neighboring weather station (NASA POWER (accessed on 8 November 2023, https://power.larc.nasa.gov/)). Considering the aforementioned factors, this study establishes a Long Short-Term Memory (LSTM) building energy prediction model based on the “Energy Detective” dataset (Shanghai, China) and data from neighboring weather station data. The aim of this study is to investigate the impact of weather errors from neighboring weather stations on building energy predictions. The main research objectives are as follows:(1)Analyze the weather data errors at the same latitude and longitude by examining the correlation between the weather variables and the target predictor variables. Additionally, analyze the impact of weather data from different source modalities at the latitude and longitude of the target building on the building’s energy prediction for the next hour and the next day;(2)Conduct a comparative analysis of the impact of weather errors from neighboring weather stations (Detective, A, B, C, D) on building energy predictions for the next hour and the next day. Analyze which weather data, when used to train the model, led to a higher prediction accuracy;(3)Consider the difference in prediction results between summer and autumn using the test dataset.

In order to facilitate a clear understanding of the work undertaken in this paper, a flowchart emphasizing the entire research process is shown in Figure 1. This paper is organized as follows: (1) Section 1 provides a detailed description of the Long Short-Term Memory Network (LSTM); (2) Section 2 describes the data preparation process in detail, which contains the description of the data and the setup of the neighboring weather station data, as detailed in Section 3.1. Among them is the data preparation process, including a description of the data and the setup of the neighboring weather station data, as outlined in Section 3.1. The data preparation process involves replacing the neighboring weather station weather data (A, B, C, D) with the original training data (Detective) for the training set and using the Detective data for the test set. Section 3.1 also includes a quantitative analysis of uncertainty, specifically the quantitative analysis of weather errors at neighboring weather stations; (3) Section 3 mainly focuses on the training and validation of the model. It includes data preprocessing, model parameter setting, and model evaluation indexes. Section 3.2 provides a detailed description of the data preprocessing, which involves two parts: Max min normalization and sliding window processing. Additionally, Section 3.4 describes the evaluation indexes of the model, including deterministic evaluation indexes and uncertainty evaluation indexes; (4) Section 4 presents the model evaluation and prediction analysis. It includes an analysis of the weather errors at the same latitude and longitude, the effect of the weather errors at adjacent weather stations on the prediction of building energy in the next hour and day, and the effect of the weather errors on the building energy in summer and the effect of the weather errors on the building energy in autumn and autumn. This section analyzes the differences between the prediction results in the summer and autumn; (5) Section 5 analyzes and summarizes the limitations of the study, as well as provides suggestions for improvements in response to weather errors at neighboring weather stations; (6) Finally, the main conclusions of the study are summarized in Section 6, along with future research directions. These studies provide new insights for improving building energy prediction modeling to enhance the efficiency and accuracy of energy management.

## 2. Theoretical Background

### 2.1. Long Short-Term Memory Networks

Long Short-Term Memory Network (LSTM) is a specialized recurrent neural network structure used for processing time series data. Compared to traditional recurrent neural networks, LSTM introduces memory units and three gates (forgetting gate, input gate, and output gate) to better capture and retain long-term dependencies [17,40]. The memory unit stores historical information, while the gating unit controls the flow of information, enabling the LSTM to selectively remember or forget specific information. Figure 2 illustrates the suitability of LSTM for time series prediction. In this study, we configured a model with multiple LSTM layers. Each LSTM layer consists of a certain number of LSTM units that transfer information through an activation function. During execution, each LSTM operation is linear. The output at each time step is computed based on the input variable $X_{k}$, the hidden layer $h_{k-1}$ (short-term memory), and the cell state $c_{k-1}$ (long-term memory). The computation process of the LSTM nerve cell is as follows: at time step k, the input value $X_{k}$ is combined with the state information $c_{k-1}$ and output value $h_{k-1}$ from time step k−1. This combination produces the state information $c_{k}$ and the output value $h_{k}$, after the effect of the forgetting gate (which eliminates useless information and retains useful information), the input gate (which updates the long-term memory), and the output gate (which reads information from the cell state $h_{k}$). The state information $c_{k}$ and output value $h_{k}$ at time step k are then used for the next cell learning computation. This process continues until the computation for all cells is completed. The specific equations for these computations are as follows:(1)$$i_{k}=\sigma (W_{i,X}\cdot X_{k}+W_{i,H}\cdot h_{k-1}+b_{i})$$
(2)$$f_{k}=\sigma (W_{f,X}\cdot X_{k}+W_{f,H}\cdot h_{k-1}+b_{f})$$
(3)$$g_{k}=tanh(W_{g,X}\cdot X_{k}+W_{g,H}\cdot h_{k-1}+b_{g})$$
(4)$$o_{k}=\sigma (W_{o,X}\cdot X_{k}+W_{o,H}\cdot h_{k-1}+b_{o})$$
(5)$$c_{k}=f_{k}\cdot c_{k-1}+i_{k}\cdot g_{k}$$
(6)$$h_{k}=o_{k}\cdot tanh(c_{k})$$
where $W_{i,X}$, $W_{i,H}$, $W_{f,X}$, $W_{f,H}$, $W_{g,X}$, $W_{g,H}$, $W_{o,X}$, and $W_{o,H}$ are the weight matrices.$\;b_{i}$, $b_{f}$, $b_{g}$ and $b_{o}$ are the bias vectors.$\;h_{k}$ represents the output of the LSTM unit at time step k. σ() denotes the Sigmoid activation function, and tanh represents the hyperbolic tangent based activation function.

For the training of the model, please refer to Section 3 for a detailed description. In this study, a training dataset was employed to train the LSTM model. We selected appropriate optimization algorithms (e.g., Adam) and loss functions to minimize the error between the prediction results and the true values. The weights and biases of the model were updated using the backpropagation algorithm. During the training process, we carefully chose suitable learning rates and batch sizes, and determined the optimal number of iterations to achieve convergence and optimal model performance. To evaluate the model’s performance, we used a validation set for model validation and tuning, thus improving the prediction accuracy by adjusting the model’s hyperparameters. Finally, we assessed the prediction ability of the final optimized model using a test set in real scenarios.

### 2.2. Predicting Next 1 h vs. Predicting Next 1 Day

In the field of building energy forecasting, predicting energy consumption for the next hour and the next day are both crucial time scales for optimizing energy management and improving energy efficiency. Forecasting building energy for the next hour enables real-time monitoring and adjustment of energy usage to meet current demand and reduce energy costs. On the other hand, forecasting building energy for the next day is essential for long-term energy planning and resource allocation. For instance, in order to optimize the operation of a thermal storage system, a forecast of the cooling load for the following day is required to make informed decisions.

In this study, we develop prediction models for both the next 1 h and the next 1 day, utilizing multiple input variables such as time information, ambient temperature, humidity, etc., to predict a single output variable, which is the building energy consumption. For the next hour prediction, we use the first 24 observations as the input to predict the next target value (next 1 h). The formula for predicting the next hour’s energy consumption is as follows:(7)$${\hat{y}}_{t+1}=\hat{f}\left(x_{t-23\sim t},y_{t-23\sim t}\right)$$
where, $x_{t-23\sim t}$ represents the input data from time moment t−23 to t (including time and weather data); $y_{t-23\sim t}$ represents the target energy from time moment t−23 to t; $\hat{f}\left(\;\right)$ represents the one-to-one mapping relationship (function) used for training the LSTM model; and ${\hat{y}}_{t+1}$ represents the predicted energy for future time t+1. Similarly, for predicting the energy consumption for the next 1 day, the formula is as follows:(8)$$\left[{\hat{y}}_{t+1},{\hat{y}}_{t+2},\ldots ,{\hat{y}}_{t+24}\right]=\hat{f}\left(x_{t-23\sim t},y_{t-23\sim t}\right)$$

During the testing phase, the LSTM employs a “direct prediction method” for the next 24 h. This means that it uses the historical data of time steps t−23, t−22, …, t all at once to predict the energy consumption for time steps t+1, t+2, …, t+24.

## 3. Experimental Setup and Data Preparation

### 3.1. Data Description and Data Setup of Neighboring Weather Stations

Herein, energy data from 15 buildings in the “Energy Detective” dataset were utilized [39]. The dataset consists of hourly energy consumption data from two-meter types in public buildings located in Shanghai, China. The energy cost of lights and plugs was one of the meter types (marked as “Q”), and that of the HVAC system was the other (marked as “W”). The energy of the HVAC system was utilized as the target variable herein. Table 1 indicates that the data utilized for analyzing the model contains time variables, as well as weather variables and energy data. Notably, the weather data for Shanghai city is included. The simulation time is generated by averaging two adjacent collection intervals, resulting in a simulation time step of 1 h.

When making energy predictions in the Shanghai area, it is necessary to consider the errors in energy consumption prediction that arise due to the difference in distance between the weather station and the target building. To investigate the effect of weather errors from neighboring weather stations on building energy prediction, this study obtained weather data for four locations (A, B, C, and D) in Shanghai from NASA POWER’s online website (https://power.larc.nasa.gov/). The specific information for each location is presented in Table 2 and displayed on the map in Figure 3. Among these locations, the Detective location represents the original weather data from the Energy Detective dataset. Location A represents the weather data downloaded from NASA POWER at the same latitude and longitude as that of the Energy Detective weather data collection site. Location D is situated in Chongming District, Shanghai, which is surrounded by the sea and experiences high weather fluctuations. Location B is near Hongqiao International Airport in Shanghai, characterized by a low building density and exhibiting some weather variations compared to other densely populated urban environments. Locations B, C, and D are situated at a distance of 25 km in different directions from Location A, thus ensuring a deviation from Location A in terms of distance.

For this study, the Energy Detective dataset serves as the original dataset. We analyzed the errors between the weather data from other weather stations (A, B, C, and D) and the Detective weather data, while disregarding the errors present in the Detective weather data itself. These errors were quantitatively and statistically analyzed using Equation (9):(9)$${Error}_{Detective-NASA}=\frac{1}{\sqrt{2{\pi \sigma }^{2}}}e^{-\frac{{\left(x-\mu \right)}^{2}}{2\sigma^{2}}}$$
where ${Error}_{Detective-NASA}$ represents the error between the original Detective weather data and the NASA weather data; μ  represents the mean of the error data, and σ represents the standard deviation of the error data. The standard deviation is a statistical measure used to assess the dispersion of data, describing the degree of dispersion of individual data values in relation to the mean. A larger standard deviation indicates higher data dispersion, and vice versa.

The uncertainty of the weather errors from neighboring weather stations varies among the stations. Each weather station has its own set of (μ, σ) values, and the coefficients of the error with respect to the original weather data are presented in Table 1. Figure 4 and Table 3 illustrate that the weather data errors generated by Equation (9) are used as the errors for the neighboring weather stations, which are then inputted into the LSTM model for training purposes. In Figure 1, the horizontal coordinate represents the error between the Detective weather data and the NASA weather data, while the vertical coordinate represents the frequency of error distribution. The graph shows that the weather errors of the neighboring weather stations for the temperature (Ta) parameter are skewed to the right (greater than 0), whereas the weather errors for the dew point temperature (Td) parameter are skewed to the left (less than 0). Additionally, the graphs indicate that for both the Ta and Td weather parameters, the errors between the Detective weather data and the A and B weather data are relatively small (1.5116 and 1.5368, respectively). Regarding the relative humidity (RH) parameter, the errors between the Detective weather data and the A and B weather data are also relatively small for all four weather stations (A, B, C, and D). Notably, for the barometric pressure (Ps) and wind speed (Sw) meteorological parameters, the errors between the Detective weather data and the four weather stations (A, B, C, and D) are similar, with errors around 0.90 for the Ps parameter and around 5.00 for the Sw parameter.

### 3.2. Data Preprocessing

In order to enhance the accuracy and reliability of the prediction results, data preprocessing techniques were employed, including normalization and sliding window processing. The specific details are outlined below:

Max min Normalization: As shown in Table 1, this study presents the input variables. It is evident that these variables exhibit different data ranges and fluctuations. In deep learning, variations in the numerical magnitude of input data can directly impact the model’s ability to extract information. Moreover, the update of the training parameters in preceding layers can alter the distribution of input data in subsequent layers. To mitigate the influence of numerical disparities on model efficacy and improve computational speed, this study applies Max min normalization [18]. The normalization equation is as follows:(10)$$x^{\prime }=\frac{x-x_{min}}{x_{max}-x_{min}}$$

In the formula, $x^{\prime }$ represents the value of a single data point, $x_{min}$ denotes the minimum value of the column containing the data, and $x_{max}$ represents the maximum value of the column containing the data.

Sliding Window Processing: Figure 5 illustrates the sliding window process using the first 24 h of historical data as input. In the initial sliding window, $x_{1}$, $x_{2}$, …, $x_{24}$ and $y_{1}$, $y_{2}$, …, $y_{24}$ are inputted into the prediction model. Here, $x_{1}$ contains the timestamp and meteorological parameters for the first moment, while $y_{1}$ represents the energy data for the first moment. These inputs are utilized to predict the energy consumption for moment 25 (i.e., predicting the next 1 h), or they can be used to simultaneously predict energy consumption for moments 25, 26, …, 48 (i.e., predicting the next 1 day). The sliding window moves forward one step at a time until all data has been processed.

### 3.3. Model Parameter Setting

In this study, a processor was utilized to calculate the energy consumption for the next 1 h and 1 day (24 h). Specifically, an AMD Ryzen 7 4800H processor with Radeon Graphics and 16 GB (3200 MHz) of memory was utilized (Computer equipment from Lenovo Group, China). Prior to experimental evaluation, hyperparameter search was conducted based on the literature [18,41]. The Grid search range for hyperparameter tuning is presented in Table 4. Grid search was employed in this study to optimize the model’s hyperparameters. For all datasets used, 70% of the data were utilized for training, while the remaining 30% were allocated for testing purposes. The optimization range of the number of neurons $n^{\left[lstm\right]}$ and the number of fully connected units $n^{\left[fc\right]}$ in the LSTM layer was set as [16, 32, 64, 128]. Additionally, the learning rate was optimized within the range of [0.01, 0.001, 0.0001, 0.00001]. The LSTM model was trained with 1000 iterations using the Adam algorithm [41,42] and an early stopping strategy was employed to ensure rapid convergence and prevent overfitting.

### 3.4. Prediction Performance Evaluation

In this study, both deterministic and uncertainty evaluation metrics were employed to assess the model [1,43,44]. The root mean square error coefficient of variation (CV-RMSE) is a scale-independent metric suitable for evaluating the performance of models constructed using different datasets [43]. The mean absolute percentage error (MAPE) is a commonly used statistical measure for prediction accuracy. It can be calculated using the following formula:(11)$$CV-RMSE=\frac{\sqrt{\frac{1}{n}\sum_{i=1}^{n}{\left({\hat{y}}_{i}-y_{i}\right)}^{2}}}{\frac{\sum_{i=1}^{n}y_{i}}{n}}$$
(12)$$MAPE=\frac{1}{n}\sum_{i=1}^{n}\left|\frac{y_{i}-{\hat{y}}_{i}}{y_{i}}\right|$$

In Equations (11) and (12), $y_{i}$ represents the actual energy consumption value, ${\hat{y}}_{i}$ denotes the predicted energy consumption value, and n represents the number of samples in the test set. The magnitude of RMSE, CV-RMSE, and MAPE all reflect the difference in model accuracy. In a given scenario, a smaller value indicates higher model accuracy. MAPE ranges from [0, +∞), where a MAPE of 0% indicates a perfect model, and a MAPE greater than 100% indicates a poor model.

The coefficient of determination $R^{2}$ (R-Square) characterizes the degree of model fit and reflects the accuracy of the model. It can be calculated using the following formula:(13)$$R^{2}=1-\frac{\sum_{i=1}^{n}{\left({\hat{y}}_{i}-y_{i}\right)}^{2}}{\sum_{i=1}^{n}{\left({\bar{y}}_{i}-y_{i}\right)}^{2}}$$

In Equation (13), ${\bar{y}}_{i}$ represents the average value of the energy consumption sample. $R^{2}$ takes the value in the range of [0, 1]. Generally, larger values of $R^{2}$ indicate better model fit.

Prediction Interval Coverage (PICP) is used to measure the percentage of observed values within the prediction interval (PI). A larger PICP indicates a greater probability that the actual value falls within the interval, thereby indicating higher prediction accurate. Prediction Interval Mean Width (PIMW) quantifies the width of the prediction interval for air conditioning energy consumption. When the PICP is fixed, a smaller PIMW suggests a narrower prediction interval and better prediction accuracy. The reliability indicator (r) represents the difference between the interval coverage and the preset confidence level. A positive value of r indicates that the model exhibits favorable bias, with reliability higher than the given confidence level. Conversely, a negative value of r indicates harmful bias, with reliability lower than the given confidence level.
(14)$$PICP=\frac{1}{n}\sum_{i=1}^{n}k_{i}^{\left(\propto \right)}$$
(15)$$PIMW=\frac{1}{n}\sum_{i=1}^{n}\left[U_{t}-L_{t}\right]$$
where n represents the sample size, and $L_{t}$ and $U_{t}$ are the lower and upper bounds of the sample PI, respectively.

## 4. Results

### 4.1. Weather Error Analysis at the Same Latitude and Longitude

Under the same latitude and longitude, there are certain errors in observing weather data, including instrumental measurement errors and human recording errors. These errors may lead to deviations between the actual observed values and the true weather conditions, which in turn can affect the results of building energy prediction. Therefore, it is important to analyze the correlation between weather data (Detective and A) and building energy at the same latitude and longitude, as shown in Figure 6. In this study, we employ the Spearman rank correlation coefficient [45] to analyze the correlation between weather data and building energy. The coefficient takes a value between −1 and 1, with −1 indicating a completely negative correlation, 1 indicating a completely positive correlation, and 0 indicating no correlation. The specific formula is as follows:(16)$$\rho =1-\frac{6\sum d_{i}^{2}}{n\left(n^{2}-1\right)}$$
where $d_{i}$ represents the difference between the bit values corresponding to the ith data; n represents the total number of observed samples.

Figure 6 displays a heat map representing the correlation coefficients of the five weather variables (horizontal coordinates: Detective and A) with the energy consumption of the 15 buildings (vertical coordinates). Positive correlations are shown in red, while negative correlations are shown in blue. The positive correlation coefficients can reach as high as 0.4294, while the negative correlation coefficients can be as low as −0.5475. Regarding the Detective weather, there is a positive correlation between temperature and wind speed with most of the building energy, while a negative correlation exists between relative humidity and the majority of the building energy. Similarly, the correlation pattern is observed in the A weather data. Comparing the Detective and A weather variables, it is found that relative humidity is more negatively correlated with energy in the A weather data compared to the Detective weather data. Overall, the correlation between A weather data and building energy is greater than that of the Detective weather data. Since the latitude and longitude of the locations of Detective and A weather are the same, the errors in Detective and A weather data cannot be neglected.

Next, the impact of Detective and A weather data on building energy prediction is analyzed, as shown in Figure 7. The box-line plots illustrate the distribution of $R^{2}$ values for prediction results obtained using Detective and A weather data in the building energy prediction training model. The horizontal axis represents the two prediction methods: predicting energy for the next hour and predicting energy for the next day (24 h). The vertical axis represents the coefficient of determination $R^{2}$, where a value closer to 1 indicates higher prediction accuracy. The white horizontal line in the graph represents the median value of the prediction result $R^{2}$. It can be observed that when training the model using Detective weather data, the median $R^{2}$ for predicting energy in the next hour and predicting energy in the next 1 day are 0.9486 and 0.8683, respectively. These values suggest that the prediction accuracy for energy in the next 1 h is higher compared to that for energy in the next 1 day. Similarly, when training the model using A weather data, the median $R^{2}$ values for predicting energy in the next hour and predicting energy in the next 1 day are 0.9509 and 0.8716, respectively. These values indicate that the prediction accuracy of this method is higher compared to the method of predicting energy in the next 1 day, and overall the $R^{2}$ values are larger. In summary, the $R^{2}$ value for predicting energy in the next hour is approximately 0.08 larger than that for predicting energy in the next day. Furthermore, when training the model using Detective and A weather data (with same latitude and longitude), the $R^{2}$ values for A weather data are more centralized, resulting in larger overall prediction results. This suggests that there may be a certain discrepancy between Detective weather data and A weather data itself. The data obtained at the latitude and longitude locations where the target buildings are located may come from different sources, leading to variations in the data.

### 4.2. The Effect of Weather Errors at Neighboring Weather Stations on Building Energy Predictions for the Next 1 h and 1 Day

This subsection analyzes the impact of weather errors from neighboring weather stations on building energy prediction for the next hour and day. Five types of neighboring weather data are considered in this study, and their latitude, longitude, and other information are presented in Table 2 and Figure 3. The violin plot of Figure 8 illustrates the distribution of CV-RMSE for building energy prediction using the five neighboring weather datasets. The horizontal axis represents the five neighboring weather datasets used for training the model, while the vertical axis represents the CV-RMSE. The figure presents two prediction methods: predicting the next 1 h energy and predicting the next 1 day energy. The short dash lines in the left-hand plot of the violin chart represent the upper quartile and lower quartile, respectively, and the dotted line is the median value. The short blue dash lines on the right side of the violin chart (Next 1 day) have the same meaning as on the left side (Next 1 h). From the distribution of CV-RMSE in Figure 8, it can be observed that the median distribution of CV-RMSE for predicting energy in the next 1 h is around 0.3, whereas the median CV-RMSE for predicting energy in the next 1 day is around 0.5. Generally, the CV-RMSE values tend to be higher when the model predicts energy patterns for the next 1 day compared to predicting patterns for the next hour.

Regarding the prediction results of the five neighboring weather datasets used for training the model in Figure 8, the median CV-RMSE for the Detective data is 0.3145 when predicting energy for the next hour. This value is relatively small compared to the other datasets. However, the distribution of the CV-RMSE for the A data used in training the model ranges from 0 to 0.6 with a more concentrated distribution. Compared to the distribution of prediction results for the Detective data, the distribution for the A data is narrower. The median CV-RMSE values for the C and D data used in training the model are relatively large, which can be attributed to their proximity to the coast and larger fluctuation in weather data. Therefore, when predicting energy for the next hour, the CV-RMSE distribution of the prediction results using the A weather data is more concentrated and smaller overall indicating better prediction performance. When the weather data used for training the model is collected from a location that deviates from the target building, the CV-RMSE distribution tends to be larger, as observed in the B, C, and D datasets. For the prediction of energy in the next 1 day, the median CV-RMSE for the Detective data used to train the model is relatively large among the five types of weather data, with a median CV-RMSE of 0.5279 and a relatively large error. The median CV-RMSE for the D data used to train the model is the largest, with a median CV-RMSE of 0.5666, which may be related to the location of the D data point. The median CV-RMSE for the C data used to train the model was 0.4810, indicating a relatively high prediction accuracy.

Figure 9 presents a Sankey plot depicting the distribution of optimal weather data types used to train the model for 15 buildings. The left side of the plot represents the 15 buildings. In the middle, there are two building energy prediction modes: predicting for the next 1 h and predicting for the next 1 day. The output on the right side shows the optimal weather data type used for training the model. For example, for Building #1, when predicting energy for the next 1 h, the highest prediction accuracy is achieved by training the model with A weather data among the five types of weather data. The same is true for predicting energy for the next 1 day. From the results obtained with the available data, it can be concluded that the fluctuation of neighboring weather data introduces uncertainty in building energy prediction results. Different weather data at the latitude and longitude of the target building may have varying spatial resolution and coverage. Some data sources may provide finer spatial resolution, while others may have wider coverage. As a result, the weather at neighboring weather stations affects the prediction error of building energy forecasts due to the presence of errors.

### 4.3. Difference in Building Energy Prediction Results between Summer and Autumn

This subsection begins with an example of Building #8 and analyzes the difference between predicting energy for the next 1 h and predicting energy for the next 1 day. The test set was used to predict energy for the next 1 h and the next 1 day, and the results are summarized in Table 5 and Figure 10. According to Table 5, in terms of deterministic prediction accuracy, the predicted MAPE value for predicting the next 1 h is 17.35%, which is 27.09% lower than the predicted MAPE value for predicting the next 1 day. The prediction model for the next 1 h outperforms the corresponding model for the next 1 day, as indicated by its lower CV-RMSE value. Figure 11 demonstrates the prediction error of energy for 1 h and 1 day. The horizontal coordinate represents the prediction error (the difference between the predicted value and the true value) while the vertical coordinate represents the frequency distribution of the energy error. From the figure, it can be observed that the prediction error for predicting energy for the next 1 h is skewed to the left, while the prediction error for predicting energy for the next 1 day is skewed to the right. Compared to the frequency distribution of prediction errors for predicting energy for the next 1 day, the frequency of distribution of prediction errors for predicting energy the next 1 h is higher, and the range of prediction errors is narrower. Figure 10 illustrates the predicted and true energy consumption distributions for the test set portion, and it is evident that the predicted model for the next 1 day has a poorer fit in the peaks and troughs compared to the predicted model for the next 1 h. This indicates that the prediction error of the next 1 day is larger than that for the next 1 h.

To account for the effect of seasonal factors, this study incorporates temporal features, such as the month and day of the week, into the energy prediction model to more accurately capture temporal variations. The prediction results of Building #8 are used to plot 95% confidence intervals, which visualize the uncertainty of the prediction results during the summer (see Figure 12) and autumn (see Figure 13) seasons based on partial data from the test set. Table 6 provides a summary of the probability metric information (e.g., PICP, PINAW, and r) for the prediction generation of the next 1 h and the next 1 day during the summer and autumn seasons. In this subsection, the prediction results for summer and autumn using partial data from the test set are analyzed to understand the differences in predictions uncertainty. Specifically, the summer is taken as 1 July 2016 0:00:00–7 July 2016 23:00:00, while the autumn is taken as 1 October 2016 0:00:00–7 October 2016 23:00:00.

According to Table 6, the prediction interval coverage (PICP) for the next 1 h of prediction during the summer is 94.64%, which is smaller than the PICP for the next 1 day of prediction. The reliability indicator (r) for the summer prediction of the next 1 h is less than 0, indicating a harmful bias in the model. Figure 12 illustrates the prediction results for the summer season. The horizontal axis represents samples, while the vertical axis represents energy consumption values. At a 95% confidence interval, the figure demonstrates that the predicted values for the next 1 h of forecasting align more closely with the true values compared to the predicted values for the next 1 day of forecasting. Notably, when predicting energy consumption 1 day in advance, the prediction bands do not show improved results at the peak.

In the autumn prediction results, the reliability index is also less than 0 for both the next hour and next day prediction models. The PICP for the next 1 h prediction is 94.64%, which is greater than the PICP for the next 1 day prediction. The prediction interval means width (PIMW) of the next 1 h model during autumn is 18.98 kWh, significantly smaller than the PIMW of the next 1 day model. Figure 13 illustrates the autumn forecast results. It is evident that the forecast band for the period from 1 October 2016 to 7 October 2016 outperforms the forecast results for the next 1 day energy consumption. Additionally, when predicting the next 1 h energy consumption, the uncertainty is lower compared to the next 1 day model (PIMW of 18.98 kWh). Furthermore, the trend of the predicted values for the next 1 h energy consumption is more similar to the true values, as shown in the figure.

In summary, the summer forecast results demonstrate a high level of accuracy for both the next 1 h and the next 1 day predictions, while the autumn forecast results exhibit higher error for the next 1 day prediction, with the next 1 h prediction achieving a higher level of accuracy. Moreover, the next 1 day forecast model has a higher PIMW value, indicating greater uncertainty compared to the next 1 h forecast model. Forecast models with shorter forecast ranges not only exhibit higher accuracy but also lower uncertainty.

## 5. Discussion

### 5.1. Contribution and Limitation

In this paper, we comparatively analyze the effect of weather data errors from five neighboring weather stations (Detective, A, B, C, and D) on energy forecasts for the next hour and day. There are some differences between the Detective and A weather data, despite their shared latitude and longitude. Among the five neighboring weather stations, the prediction accuracy for next 1 h energy consumption is higher compared to the prediction accuracy for the next 1 day pattern. This study provides readers with insights to consider weather errors from neighboring weather stations.

This study addresses energy prediction for buildings, which has a wide range of applications in improving building efficiency, reducing operating costs, and enhancing building–grid interactions. The main contribution of this study is twofold. Firstly, it considers the errors in weather data from adjacent weather stations used for energy prediction. Secondly, it analyzes two energy prediction models: next 1 h and next 1 day. The limitations and challenges of this study are as follows:(1)Regional weather error modeling: We examine the impact of weather errors from neighboring weather stations on building energy forecasts. However, accurately modeling weather errors poses certain challenges. The weather data we use is limited to a specific time frame and geographical area, which may not encompass all variations. For instance, when the distance from point A is 5 km, weather data different from point A will not be retrieved from the NASA POWER online weather data, which restricts the handling of regional weather variations.(2)Model Complexity: In our study, we employ the LSTM model as the base model for building energy prediction (choosing the appropriate model itself is a challenge, as different models may yield varying results). However, the LSTM model has several parameters that require tuning, such as hidden layer size, learning rate, etc. We face challenges in selecting model parameters, as different choices can impact prediction performance. Therefore, careful adjustment and optimization of model parameters are necessary.(3)Geographical differences and applicability: Our study conducts experiments based on data from specific regions and buildings. While we provide readers with insights into considering weather errors from neighboring weather stations, there are geographic differences and variations in energy consumption characteristics across different regions and buildings. Hence, our findings require further validation and adaptation analysis before they can be applied to other regions and buildings.

### 5.2. Recommendations for Improvement

This subsection summarizes some improvement suggestions for weather data errors from neighboring weather stations and their impact on energy prediction. The suggestions mainly focus on improving the quality control of weather data, increasing the density of weather stations, the fusion of multi-source data, model integration, and integrated prediction, as follows:(1)Improvement of meteorological data quality control: To enhance the accuracy of weather data, it is recommended that meteorological stations implement robust data quality control measures. This includes regular calibration and maintenance of meteorological measurement equipment to ensure data accuracy and consistency. Additionally, establishing a sound data collection and recording mechanism is advised to ensure data integrity and reliability.(2)Increase the density of weather stations: To mitigate the impact of weather errors from neighboring weather stations on building energy forecasts, it is recommended to augment the density of weather stations. By deploying weather stations in more geographical locations, weather changes in different areas can be better captured, ultimately reducing the influence of weather errors.(3)Fusion of multi-source data: In addition to relying on data from a single weather station, it is recommended to leverage multi-source data fusion. For instance, integrating multiple data sources such as satellite data, radar data, and sensor networks can provide a more comprehensive and accurate understanding of weather conditions. Multi-source data fusion can enhance the comprehension and prediction of weather changes, thus improving the accuracy of building energy forecasts.(4)Model integration and integrated prediction: It is advised to employ model integration to enhance the accuracy and robustness of energy prediction. By combining predictions from multiple models, the bias and uncertainty inherent in a single model can be mitigated. Additionally, integrated forecasting can be considered to incorporate the uncertainty of meteorological data into energy forecasting, thereby providing more reliable results.

## 6. Conclusions and Outlook

Building energy prediction is of great importance in energy management and energy conservation, particularly in the fields of thermal storage operation and smart grid management, which have wide-ranging applications. Weather variables play a crucial role in building energy prediction. In order to analyze the impact of weather data errors on building energy prediction, this study first evaluates the errors in weather data and analyzes the effect of weather data from different sources, considering the latitude and longitude of the target building on the prediction of building energy for the next 1 h and next 1 day. Subsequently, the study examines the influence of weather data errors from neighboring weather stations (Detective, A, B, C, and D) on building energy prediction. Finally, the study considers the impact of data from the test set on the prediction results, specifically comparing the differences between summer and autumn. The conclusions drawn from this analysis are as follows:(1)At the same latitude and longitude, the correlation between the A weather data and the target building energy is higher compared to the Detective weather data. This suggests that there are some differences between the Detective and A weather data at the same latitude and longitude.(2)When considering the five neighboring weather stations, the prediction accuracy for the next 1 h energy consumption is higher compared to the prediction accuracy for the next 1 day energy consumption. The median $R^{2}$ value for the next 1 h reaches 0.95, which is approximately 0.08 higher than the $R^{2}$ value for the next 1 day.(3)In terms of seasonal differences, the summer season achieves higher predictions for both the next 1 h and the next 1 day energy consumption, while the autumn season achieves higher predictions only for the next 1 h energy consumption. The 1 day prediction model exhibits higher PIMW values (139.0 in summer and 146.1 in autumn) and greater uncertainty compared to the 1 h prediction model.

Considering the incompleteness and errors in weather data, future research could focus on improving the modeling and handling of uncertainty. Probabilistic models could be considered to provide estimates of the probability distribution of forecast outcomes, enabling a more accurate reflection of the impact of incomplete data on forecasts. Additionally, future research could explore methods for selecting and optimizing input features to enhance the performance of predictive models. Feature selection algorithms, such as statistical or information-theoretic based methods, can be employed to automatically filter and select the most relevant features, thereby minimizing the impact of incomplete data on the predictive model. Furthermore, data fusion, model optimization, and optimization of weather station layout can be explored to address the impact of weather data errors from neighboring weather stations on building energy forecasts. In practice, building predictive control input data often require precise temporal resolution (e.g., 20 min or less) and shorter prediction times, whereas weather data are typically provided at coarser temporal resolution (e.g., hourly or daily). Moreover, weather variables such as solar irradiance, humidity, and precipitation are highly correlated, which can introduce significant uncertainty in weather forecasts, subsequently affecting energy forecasts.

## Figures and Tables

**Figure 1 sensors-24-01157-f001:**
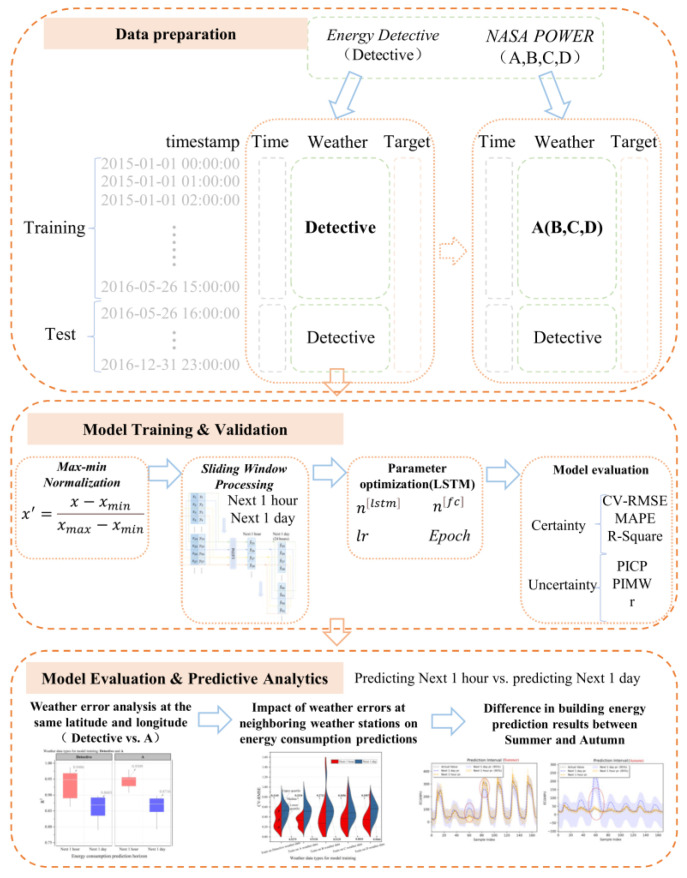
Research framework diagram for this study.

**Figure 2 sensors-24-01157-f002:**
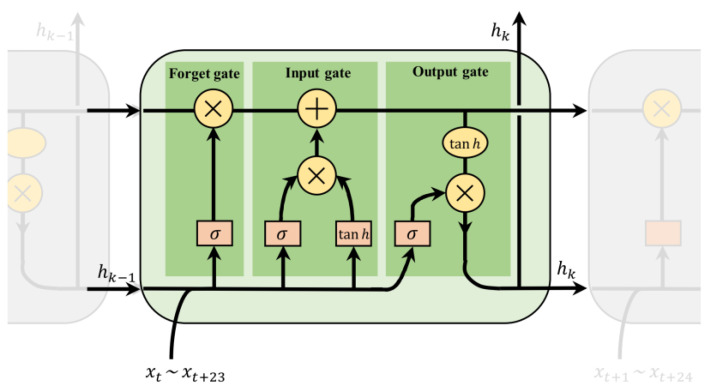
Schematic diagram of LSTM structure.

**Figure 3 sensors-24-01157-f003:**
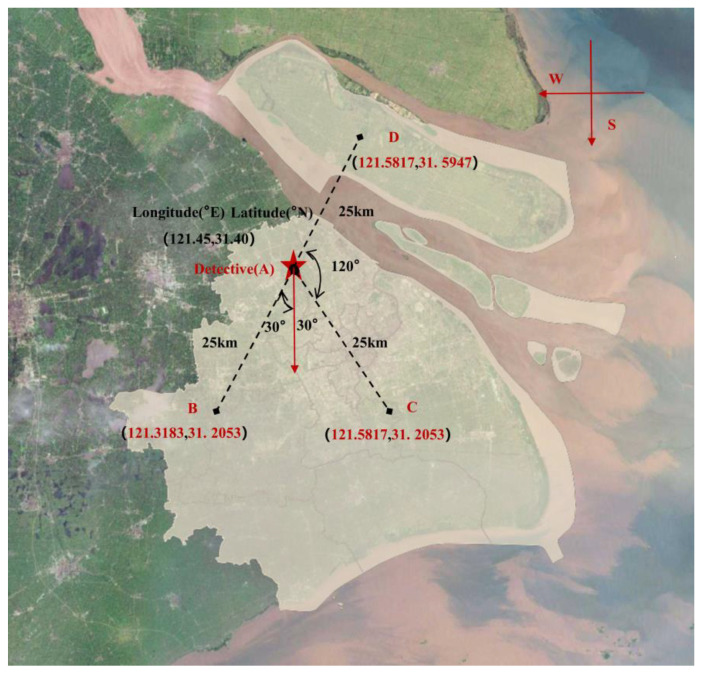
Neighboring weather station data setup location information map (location approximate only).

**Figure 4 sensors-24-01157-f004:**
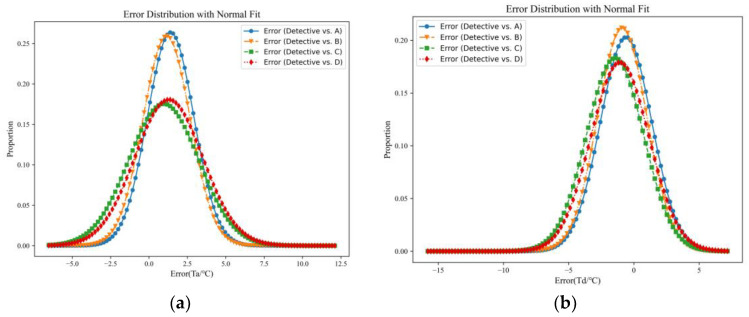
Distribution of weather errors for neighboring weather stations: Detective vs. NASA POWER ((**a**) Distribution of errors for Ta; (**b**) Distribution of errors for Td; (**c**) Distribution of errors for RH; (**d**) Distribution of errors for Ps; (**e**) Distribution of errors for Sw).

**Figure 5 sensors-24-01157-f005:**
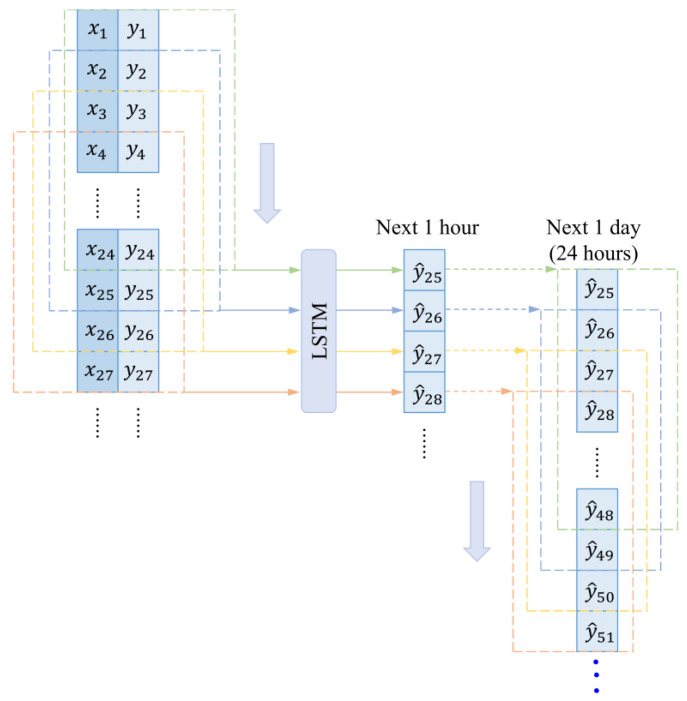
Schematic diagram of the sliding window process.

**Figure 6 sensors-24-01157-f006:**
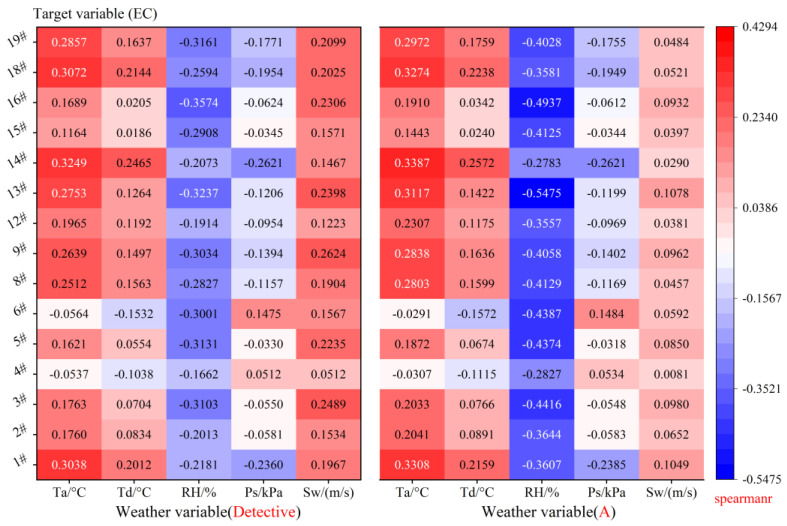
Heat map of Spearman rank correlation between weather (Detective and A) variables and building energy.

**Figure 7 sensors-24-01157-f007:**
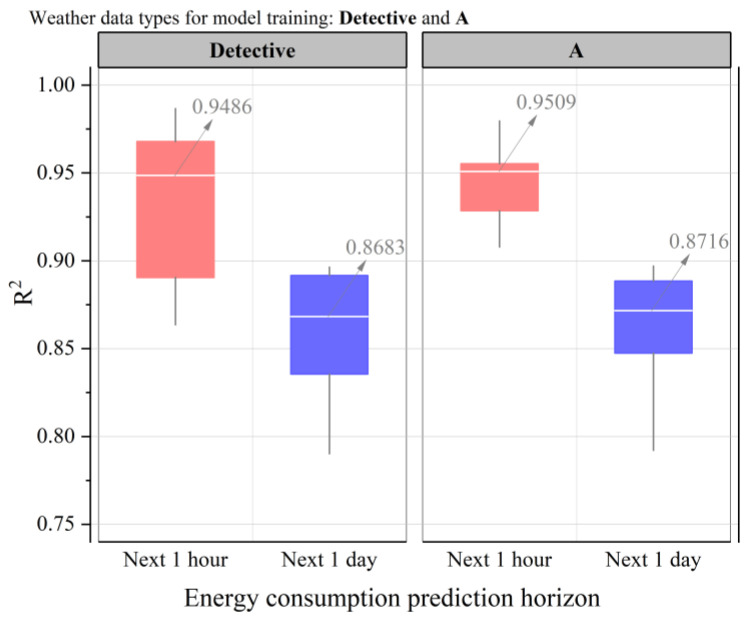
Impact of same latitude and longitude weather data used to train models on building energy prediction: Detective and A.

**Figure 8 sensors-24-01157-f008:**
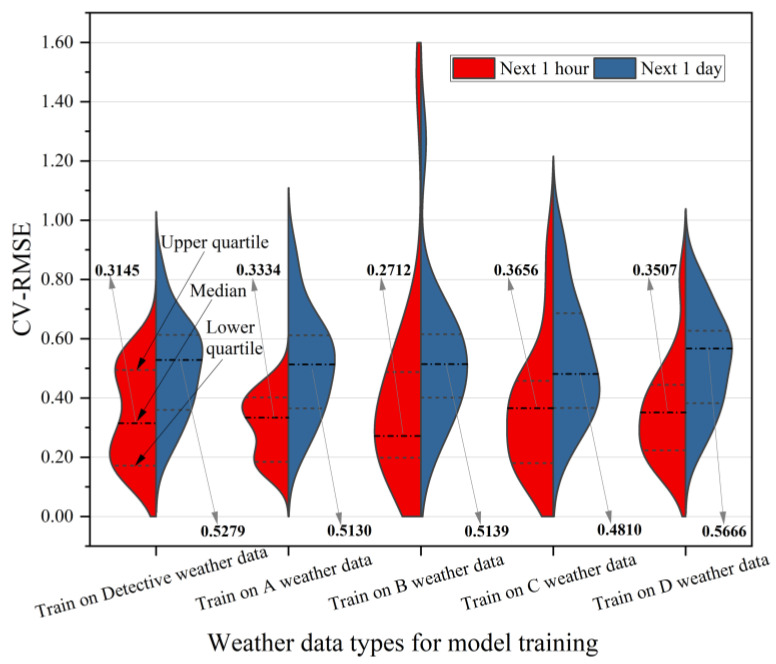
Impact of neighborhood weather data used to train models on building energy predictions.

**Figure 9 sensors-24-01157-f009:**
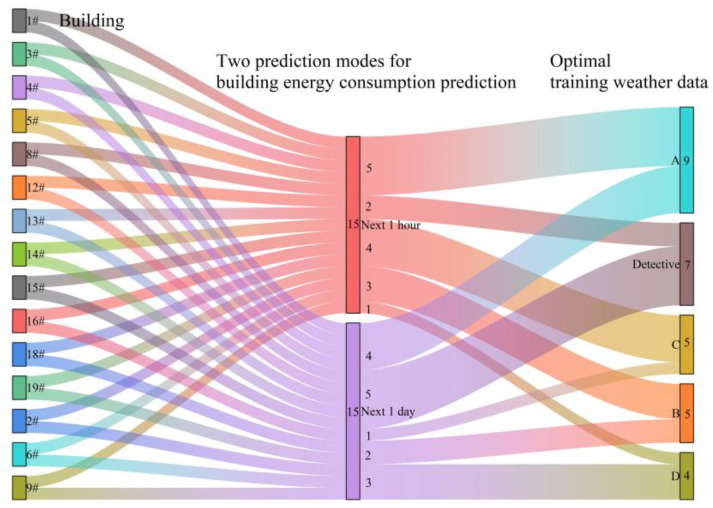
Distribution of weather data types for optimally trained models under different buildings: next 1 h and next 1 day.

**Figure 10 sensors-24-01157-f010:**
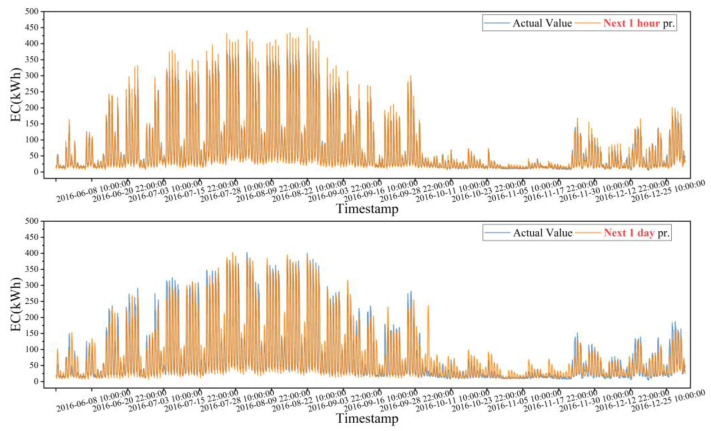
Distribution of true and predicted values for summer and autumn: next 1 h and next 1 day.

**Figure 11 sensors-24-01157-f011:**
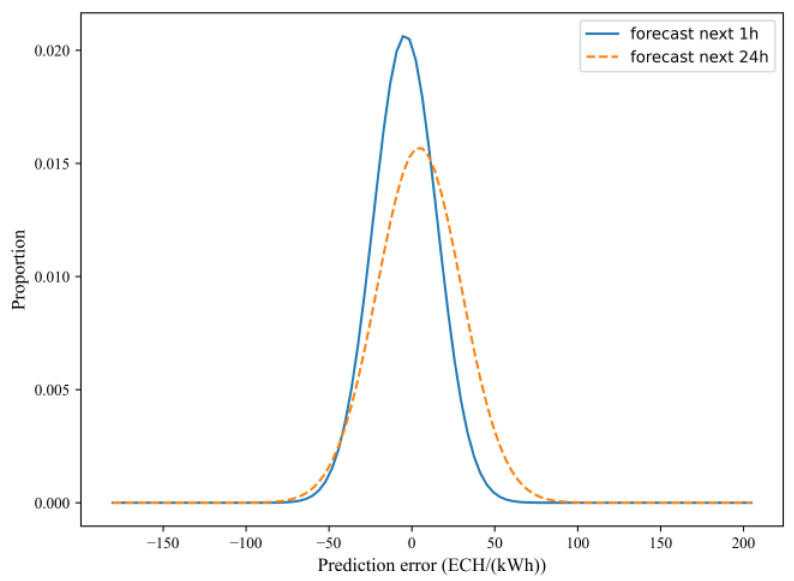
Prediction error of energy for the next 1 h and the next 1 day of forecasting.

**Figure 12 sensors-24-01157-f012:**
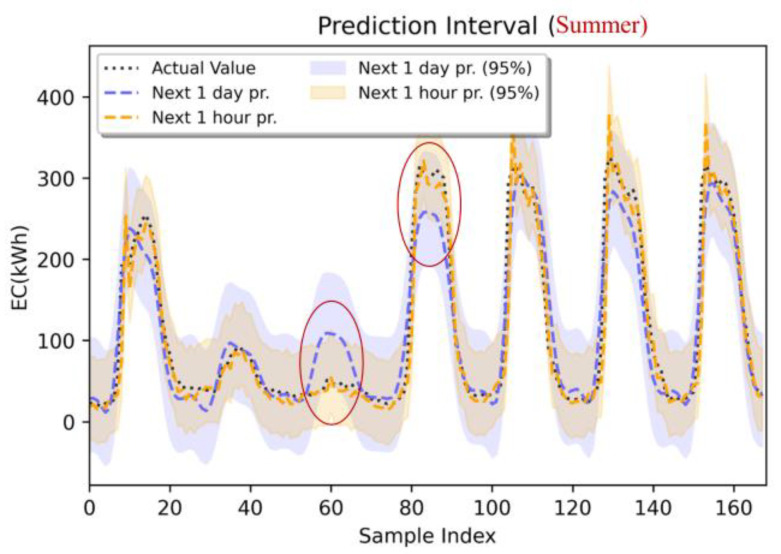
Prediction (LSTM: next 1 h, next 1 day), 95% prediction interval, actual value: summer (1 July 2016 0:00:00–7 July 2016 23:00:00).

**Figure 13 sensors-24-01157-f013:**
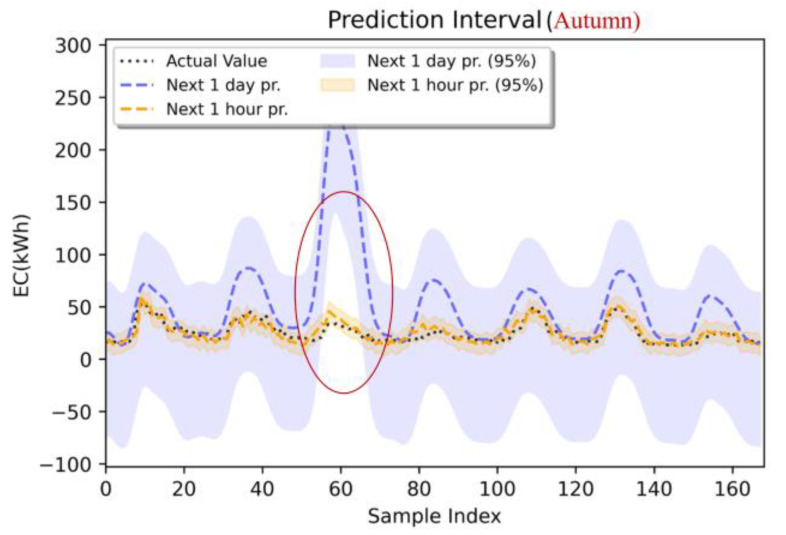
Prediction (LSTM: next 1 h, next 1 day), 95% prediction interval, actual value: autumn (1 October 2016 0:00:00–7 October 2016 23:00:00).

**Table 1 sensors-24-01157-t001:** Input variable categories.

Classifications	Details
Calendar Information	Each month of the year (m), day of the week (wk: 1, 2, …, 7), each hour of the day (h)
Weather variables	Temperature ($T_{a}$/°C), Dew point temperature ($T_{d}$/°C), Relative humidity (RH/%), Atmospheric pressure ($P_{s}$/kPa) and Wind speed ($S_{w}$/(m/s))
Target Variables	Hourly electric consumption (${EC}_{h}/(kWh)$)

**Table 2 sensors-24-01157-t002:** Weather Data Sources Information Sheet.

Data Sources	Status	Longitude (°E)	Latitude (°N)	Description
Energy Detective	Detective	121.45	31.40	Energy Detective Dataset
NASA POWER	A	121.45	31.40	Energy Detective Weather Data Collection Service
	B	121.3183	31.2053	25 km in the direction of 30 degrees south-west of A
	C	121.5817	31.2053	25 km in the direction of 30 degrees south-east of A
	D	121.5817	31.5947	25 km in the direction of 30 degrees north-east of A

**Table 3 sensors-24-01157-t003:** Error statistics for weather errors at neighboring weather stations.

Error Type		Ta	Td	RH	Ps	Sw
${Error}_{Detective-A}$	μ	1.4088	−0.5998	−8.2081	0.6099	9.0944
	σ	1.5116	1.9653	11.2654	0.9107	5.0116
${Error}_{Detective-B}$	μ	1.1496	−0.8968	−8.5196	0.6303	9.1922
	σ	1.5368	1.8825	10.9064	0.9098	5.0003
${Error}_{Detective-C}$	μ	0.9618	−1.4439	−9.6293	0.6003	8.4076
	σ	2.2727	2.1766	12.2022	0.8967	5.0007
${Error}_{Detective-D}$	μ	1.2894	−1.0714	−9.3629	0.5830	8.3859
	σ	2.2086	2.2270	12.4564	0.8970	4.9593

**Table 4 sensors-24-01157-t004:** Hyperparameters of Grid search and its search space.

Hyperparameters	Descriptions	Search Space
$n^{\left[lstm\right]}$	Number of neurons in the LSTM layer	[16, 32, 64, 128]
$n^{\left[fc\right]}$	Number of neurons in the fully connected layer	[16, 32, 64, 128]
lr	Step size shrinkage used in update to prevent overfitting	[0.01, 0.001, 0.0001, 0.00001]
Epoch	The number of times the model is fully trained using all data from the training set	1000
Optimizer	Optimization method	Adam [41]

**Table 5 sensors-24-01157-t005:** Deterministic metric prediction results for Building #8.

Evaluating Indicator	Next 1 h	Next 1 Day
CV-RMSE	0.2805	0.4080
MAPE	17.35%	44.44%
$R^{2}$	0.9476	0.8890

**Table 6 sensors-24-01157-t006:** Probabilistic metric prediction results for Building #8.

Evaluating Indicator	Title 2		Title 3	
	Next 1 h	Next 1 Day	Next 1 h	Next 1 day
PICP	94.64%	96.43%	94.64%	93.45%
PIMW	102.1	139.0	18.98	146.1
r	−0.0036	0.0143	−0.0036	−0.1555

## Data Availability

Data are contained within the article.

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
