# Peer review of "Study on the Impact of Building Energy Predictions Considering Weather Errors of Neighboring Weather Stations"

_sensors, 2024, doi:10.3390/s24041157_

Round 1
Reviewer 1 Report
Comments and Suggestions for Authors
Dear authors, this is a great paper, almost no modifications to make.
Best regards,

Reviewer 2 Report
Comments and Suggestions for Authors
Dear Authors, the paper in general is very good and bring a great value to the field of estimation and forecasting buildings energy demands in function of weather conditions. Great work :)
Below you will find little errors that should be corrected:
1. Line 202 - no space between utilized and [21]
2. Equation 6 - is there a 6 (number six) before sum in the fraction part ?
3. Lines 462 and 463 - check if there should be 1 hour in line 462 - please comment.
Reviewer 3 Report
Comments and Suggestions for Authors
1.Weather variables are very important for building energy, this paper discusses the effect of weather data from different source modes under the latitude and longitude for the target building on the prediction of building energy including the effect of weather error from neighboring weather stations on the prediction of building energy. It is suggested to conduct a quantitative analysis of the impact, especially from a practical application perspective.
2.The article begins with an introduction to uncertainty, but there is no quantitative analysis of uncertainty in the text.
3.In the second paragraph of 1.3, the research content of this article is directly introduced, and it is recommended to focus on the introduction in section 1.4.
4.Why choose the method proposed in this article? What are the shortcomings of other methods? What are the advantages of this adopted LSTM method? Is there any improvement in the methodology for this study?
5.The citation format is incorrect, and the citation cannot be displayed correctly in the main text. Please update all reference citations.
6.Please standardize the reference format.
Comments on the Quality of English LanguageFurther English polishing is possible.
Reviewer 4 Report
Comments and Suggestions for Authors
This study focuses on how errors in weather data from various sources can impact the accuracy of building energy predictions. A long-short-term memory (LSTM) prediction model was developed using the "Energy Detective" dataset from Shanghai, China, and data from neighboring weather stations. The study involves:
Analyzing errors from different weather data sources (Detective and A) located at the same latitude and longitude.
Investigating the impact of weather data errors from multiple neighboring weather stations (Detective, A, B, C, and D) on energy forecasts for the next 1 hour and 1 day, including the selection process of these weather stations.
Comparing summer and autumn forecast results.
Key findings include a significant correlation between weather data errors and energy consumption predictions. The model's median R-Square for 1-hour predictions reached 0.95, indicating high accuracy. However, predictions for the next day showed greater uncertainty, with a higher prediction interval mean width (139.0 in summer and 146.1 in autumn).
The article presents intriguing insights; however, I believe it would benefit from addressing the following points before publication:
Methodological Clarity: Provide a more detailed explanation of the LSTM model and its configuration for those unfamiliar with this approach.
Data Source Justification: Explain why the specific weather stations (Detective, A, B, C, and D) were chosen and how they represent the wider area.
Error Analysis Depth: Expand on the nature and types of errors encountered in the weather data and their direct impact on predictions.
Seasonal Variability: Elaborate on why there are differences in prediction interval mean width between summer and autumn.
Graphical Representations: Include graphical representations of the predictions versus actual data for visual clarity.
Comparative Analysis: Compare the performance of your LSTM model with other existing prediction models, if any.
Limitations and Challenges: Discuss the limitations of your study and challenges faced during the research.
Practical Implications: Highlight the practical implications of your findings for energy management in buildings.
Recommendations for Improvement: Offer recommendations for improving weather data accuracy and its impact on energy prediction.
Future Research Directions: Suggest areas for future research based on your study’s findings, particularly in improving prediction models with imperfect data.
Round 2
Reviewer 4 Report
Comments and Suggestions for Authors
The authors have revised the manuscript to a standard that I deem suitable for publication.